# Effects and cost-effectiveness of postoperative oral analgesics for additional postoperative pain relief in children and adolescents undergoing dental treatment: Health technology assessment including a systematic review

Henrik Berlin[1,2]*, Martina Vall[3], Elisabeth Bergenäs[3], Karin Ridell[1], Susanne Brogårdh-Roth[1], Elisabeth Lager[1], Thomas List[4], Thomas Davidson[2,5], Gunilla Klingberg[1,2]

1 Department of Pediatric Dentistry, Faculty of Odontology, Malmö University, Malmö, Sweden, 2 Health Technology Assessment—Odontology (HTA-O), Faculty of Odontology, Malmö University, Malmö, Sweden, 3 Malmö University Library, Malmö University, Malmö, Sweden, 4 Department of Orofacial Pain and Jaw Function, Faculty of Odontology, Malmö University, Malmö, Sweden, 5 Department of Medical and Health Sciences (IMH), Linköping University, Linköping, Sweden

* henrik.berlin@mau.se

## Abstract

### Background

There is an uncertainty regarding how to optimally prevent and/or reduce pain after dental treatment on children and adolescents.

### Aim

To conduct a systematic review (SR) and health technology assessment (HTA) of oral analgesics administered after dental treatment to prevent postoperative pain in children and adolescents aged 3–19 years.

### Design

A PICO-protocol was constructed and registered in PROSPERO (CRD42017075589). Searches were conducted in PubMed, Cochrane, Scopus, Cinahl, and EMBASE, November 2018. The researchers (reading in pairs) assessed identified studies independently, according to the defined inclusion and exclusion criteria, following the PRISMA-statement.

### Results

3,963 scientific papers were identified, whereof 216 read in full text. None met the inclusion criteria, leading to an empty SR. Ethical issues were identified related to the recognized knowledge gap in terms of challenges to conduct studies that are well-designed from methodological as well as ethical perspectives.

**Data Availability Statement:** All relevant data are within the manuscript and its Supporting Information files.

**Funding:** This study was supported by research funds from Oral Health Related Research by Region Skåne (Odontologisk Forskning i Region Skåne, OFRS 569491), Sweden. The funders had no role in study design, data collection and analysis, decision to publish, or preparation of the manuscript The funders had no role in study design, data collection and analysis, decision to publish, or preparation of the manuscript.

**Competing interests:** The authors have declared that no competing interests exist.

## Conclusions

There is no scientific support for the use or rejection of oral analgesics administered after dental treatment in order to prevent or reduce postoperative pain in children and adolescents. Thus, no guidelines can be formulated on this issue based solely on scientific evidence. Well-designed studies on how to prevent pain from developing after dental treatment in children and adolescents is urgently needed.

## Introduction

Many patients associate dental treatment with pain. There are several reasons for this, and depending on the underlying diagnosis and type of treatment, the risk of pain is realistic and should be tackled. This is especially important in children and adolescents, as they may be more vulnerable to pain owing to their level of cognitive reasoning and understanding [1]. Painful medical/dental episodes, along with minor everyday pain experiences such as bumps, falls etc., are also likely to play a significant role in shaping the individual's pain perception in future medical and/or dental events [2]. Furthermore, painful dental treatment experiences have been identified as essential components in the development of dental fear and anxiety [3, 4], which affects approximately 9% of the paediatric population [3]. Therefore, preventing and reducing pain are major responsibilities for the dental team.

Apart from using local anaesthetics, administration of oral analgesics might be one way to prevent dental treatment pain and probably even more so during the postoperative period: after tooth extractions, for example. However, there is an uncertainty regarding the use of oral analgesics in paediatric dentistry [5] and a need for more general strategies. Before constructing guidelines for this purpose, the effects and cost-effectiveness of oral analgesics as well as the ethical aspects of the intervention should be scientifically evaluated, implying a need for a health technology assessment (HTA) as well as a systematic review (SR) [6, 7].

A recent systematic review of preoperative administration of oral analgesics could not determine whether this administration is of any benefit for children and adolescents undergoing dental treatment under local anaesthetic [8]. There is, so far, no available systematic review of postoperative administration of oral analgesics in conjunction with dental treatment in children. PROSPERO (available at https://www.crd.york.ac.uk/prospero/) has no information on published or ongoing review registered on this topic other than the present study.

This HTA and SR aimed to assess the effects, adverse events, and cost-effectiveness of oral analgesics given immediately after dental treatment in order to prevent and/or reduce postoperative pain in children and adolescents aged 3–19 years. The review also sought to assess the ethical aspects of the intervention.

## Materials and methods

### Inclusion criteria

The following research questions were addressed:

- Which is the most effective (most pain-reducing as measured by a pain rating scale) oral analgesics, administered after dental treatment, in order to prevent or alleviate postoperative pain after dental treatment in children and adolescents aged 3–19 years?

- Is any pharmacological substance superior regarding preventing/alleviating pain? Is a single-dose sufficient or does a several dosage regimen have better effect?

- Are there any side effects or adverse reactions reported when oral analgesics are administered immediately after dental treatment in children and adolescents aged 3–19 years?

- Are oral analgesics given after dental treatment considered cost-effective in children and adolescents aged 3–19 years?

  A PICO model was constructed (*participants*, *interventions*, *control*, and *outcome*):
  *Participants*

- Children and adolescents aged 3–19 years

  *Interventions*

- Administration of oral analgesics after dental treatment

  ○ Pharmacological substances: prescription-free/over-the-counter oral analgesics containing paracetamol (acetaminophen), ibuprofen, diclofenac or naproxen

  ○ Administration of drug: oral administration as a single dose or multiple doses following an administration regimen

  ○ Dental treatments: primary or permanent teeth treated by filling therapy, pulp therapy/capping, tooth extraction, minor oral surgery

  *Control*

- Postoperative administration of other oral analgesics or placebo after same dental treatment

  ○ Pharmacological substances: other prescription-free/over-the-counter oral analgesics containing paracetamol (acetaminophen), ibuprofen, diclofenac, or naproxen, or placebo or no control

  ○ Administration of drug: oral administration as a single dose or multiple doses following an administration regimen

  ○ Dental treatments: primary or permanent teeth treated by filling therapy, pulp therapy/capping, tooth extraction, minor oral surgery

  *Outcome measures*

- Pain after dental treatment assessed by the child patient using Visual Analog Scale (VAS) [9], Faces Pain Scale–Revised [10], Wong-Baker FACES® [11], Numerical Rating Scale [12], Eland Color Scale [13], or other facial scales

- Adverse effects, side effects

- Costs, cost-effectiveness

  *Types of studies*

- Randomized control trials (RCT), systematic reviews (not narrative), observational studies, studies using qualitative methods

## Exclusion criteria

- Participants 20 years or older; studies where data could not be extracted for 3–19-year-olds

- Disability or medical conditions leading to cognitive impairment or neuropsychiatric diagnosis

- Oral analgesics other than paracetamol (acetaminophen), ibuprofen, diclofenac, or naproxen, or routes of administration other than *per os*

- Treatment under hypnosis, sedation, or general anaesthesia

- Pain assessment by proxy

- Languages other than English, Swedish, Danish, or Norwegian

## Literature search strategy

The protocol for this systematic review (SR) and health technology assessment (HTA) was registered on PROSPERO (CRD42017075589), September 1, 2017, available at http://www.crd.york.ac.uk/PROSPERO/display_record.php?ID=CRD42017075589.

Studies were identified using PubMed via NML, Cochrane via Wiley, Scopus via Elsevier, CINAHL via EBSCO and Embase via Elsevier. The literature searches were conducted in November 2017 and updated on November 20–23, 2018. Search strategies are presented in S1 File. Limitations were set to randomized control studies, systematic reviews (not narrative), observational studies, studies using qualitative methods, and publication year 1980 or later. There were no language restrictions. The literature search was done together with librarians specialized in informatics at the Malmö University library. Table 1 presents the number of articles identified via each database. After duplication control and removing articles published earlier than 1980, a total of 3,963 studies were finally evaluated according to the framework of the PRISMA-statement [14]. The number of abstracts retrieved, included and excluded articles, and the stage of exclusion are shown in a flowchart (Fig 1). No search for grey literature was performed.

All abstracts were screened independently by the review authors reading in pairs, according to the defined inclusion and exclusion criteria. If at least one reviewer considered an abstract relevant, the paper was included and read in full text.

**Table 1. Results from each database search.**

| Database | Date | Number of articles |
|---|---|---|
| PubMed via NLM | 20th November | 1216 |
| Cochrane via Wiley | 20th November | 1760 |
| Scopus via Elsevier | 21st November | 1972 |
| CINAHL via EBSCO | 23rd November | 469 |
| Embase via Elsevier | 23rd November | 2586 |
| | Total | 8003 |
| | Duplicate articles or published before 1980 | 4040 |
| | Number of articles evaluated according to PRISMA-statement | **3963** |

Number of articles identified via each database after updated search November 2018.

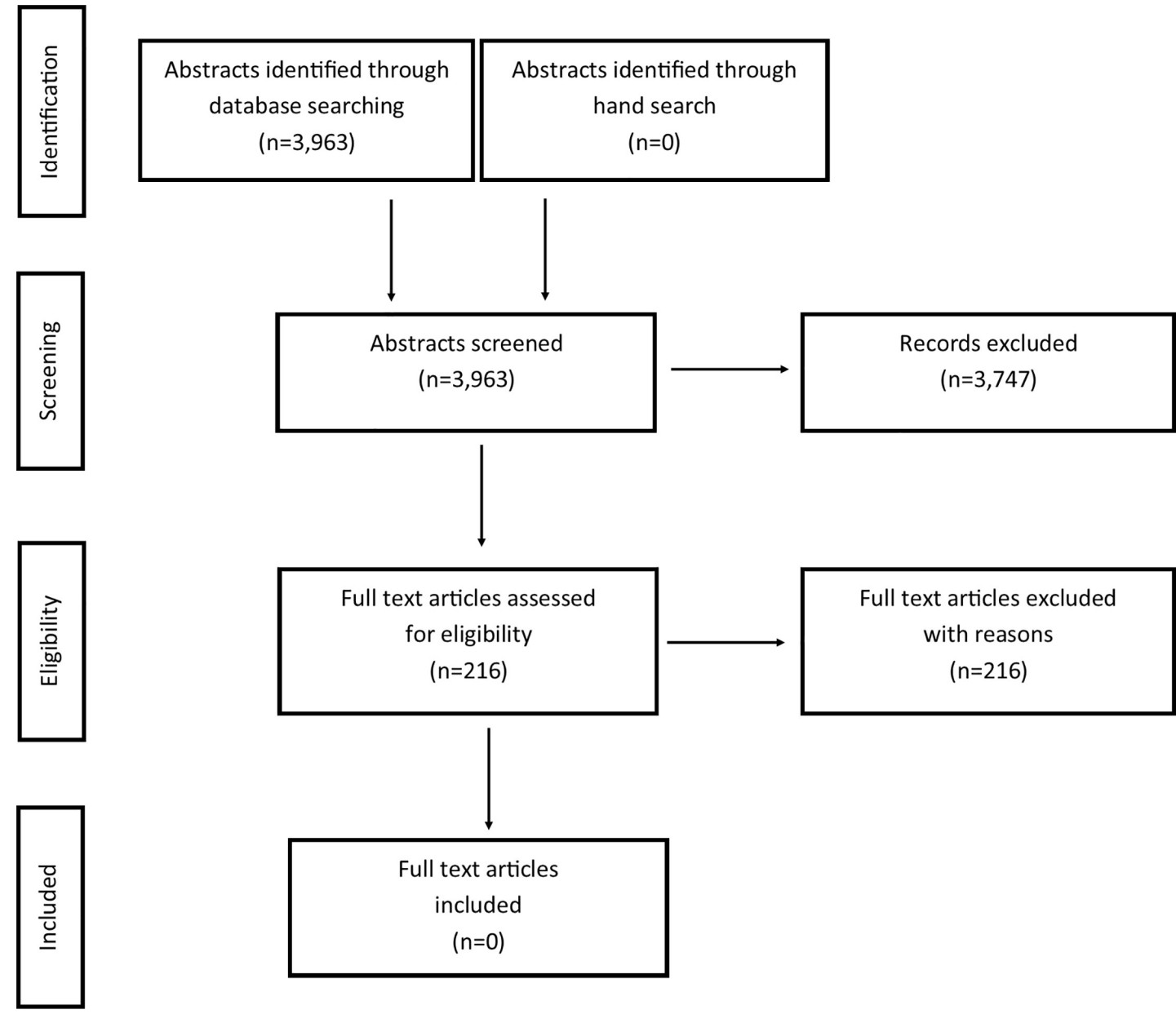

**Fig 1. Flow diagram showing the literature review process.**

## Data extraction and quality assessment

The review authors, using the same pairs as when screening abstracts, assessed the relevance of the included full-text papers. The articles were assessed independently, and any differences were handled with discussion to arrive at a consensus within each review pair. Excluded full-text papers are shown in S2 File. The following steps were also planned for risk of bias assessment, data extraction, and grading of quality.

- For assessment of relevance and risk of bias: the standardized checklist from Swedish Agency for Health Technology Assessment and Assessment of Social Services (SBU), which is similar to the Cochrane checklist (http://www.cochrane.org/) but has additional items [15, 16].

- For grading quality of evidence for studies with low or moderate risk of bias: the Grading of Recommendations Assessment, Development and Evaluation (GRADE) system [17].

- For assessment of systematic reviews: A MeaSurement Tool to Assess systematic Reviews (AMSTAR) [18].

## Results

### Literature search

The combined search from the five different databases resulted in 3,963 papers, of which 3,747 studies were excluded after reading titles and abstracts. The remaining 216 studies were retrieved and read in full text. No study was found to meet the criteria for inclusion (see Fig 1). Thus, no quality assessment or further analyses of the effects or cost-effectiveness of postoperative oral analgesics were made. S2 File, shows full references and reasons for exclusion of the 216 studies. Common reasons for exclusion were: no data for 3–19-year-olds, other pharmacologic substances (e.g. narcotic analgesics, agents not available over-the-counter), or patients treated under general anaesthesia or under sedation (e.g. benzodiazepine or nitrous oxide/oxygen sedation).

### Complications and side effects

Complications and side effects were evaluated during all stages of data extraction: i.e. also based on abstracts for papers that were not read in full text. There were no serious side effects or adverse effects reported in the retrieved full-text papers or in any of the assessed abstracts.

### Ethics

The present systematic review was unable to identify and include studies for quality assessment, and thus ethical aspects were considered on a general level, based on the framework by Heintz et al. [7]. The main ethical issues concerns the theme *effects on health* and the item *knowledge gap* (if there is insufficient scientific evidence for an intervention, are there any ethical or methodological issues for further research?) and the theme *compatibility with ethical norms* and the item *autonomy* (are the patients able to consent to the intervention?) [7]. Compliance with ethical standards was not evaluated in any of the identified studies, but some general thoughts can still be pinpointed. Explicit protections to safeguard children's rights and welfare are always necessary in medical and dental treatment, as they are in all research involving young individuals [19]. A knowledge gap as identified in the present study, signals a need for more studies. However, before including children in research on the effect of oral analgesics, well-designed studies in adults should be identified and scrutinized. The present study did not investigate this. Still, the perspective of children should be acknowledged, and children and adolescents must not be excluded from research that can be beneficial for them.

## Discussion

This health technology assessment (HTA) and systematic review (SR) was performed to assess the effects, adverse or side effects, and cost-effectiveness of oral analgesics administered to children and adolescents after dental treatment to prevent postoperative pain. This is an intervention commonly used in clinical paediatric dentistry that has previously not been systematically evaluated. As no studies meeting the inclusion criteria could be identified, it was not possible to find scientific support for the effects of postoperatively administered oral analgesics for the

prevention or reduction of pain after dental treatment in children and adolescents. Thus, this remains a knowledge gap. Based on the identified studies this HTA and SR could not identify any severe adverse events or side effects of over-the-counter oral analgesics. However, as the published literature does not provide support for the use or rejection of postoperative administration of oral analgesics in dental care for children and adolescents, it is not possible to formulate clinical guidelines on this issue solely based on scientific evidence.

In order to provide a basis for guidelines and to bridge research with decision-making, this SR was expanded to also be an HTA [6, 20, 21]. HTA includes evaluation of both ethical aspects and health economics and is an important tool when reviewing scientific evidence in order to appraise how the value of scientifically based knowledge can be implemented in health care systems and society more broadly [6, 20].

As no studies were found, we do not know the effects or the cost-effectiveness of oral analgesics given immediately after dental treatment in children and adolescents aged 3–19 years. However, decisions on this issue are continuously being made every time a child or adolescent undergoes dental treatment, so in the absence of evidence, it is important to consider other types of knowledge [22]. It is therefore important to consider the most realistic consequences of the different alternatives. The direct cost of oral analgesics is considered low, so the cost-effectiveness of the methods depends heavily on the effect side. In this situation, the decision to use oral analgesics, as well as the type of substance and whether to use a single dose or multiple dose regimen, should primarily depend on the clinical effects (including side effects) and not their cost-effectiveness. However, if it is proven that oral analgesics do not provide any additional effect, they should not be considered cost-effective. In future studies of treatment of dental pain in the child and adolescent population, it would be of value also to estimate their cost-effectiveness in order to guide decision makers in their prioritization process.

Regarding ethical aspects, the first choice should be to answer research questions by performing clinical trials in adults. However, it may be unethical to not involve children in research studies evaluating drugs. If children were excluded from all drug research, medication used in children would be limited to extrapolation from adult studies or even exclude children from the possibility of receiving existing and new drugs that they could benefit from. Thus, the research community has a significant responsibility to design, approve, and conduct high-quality studies in children so that they can have access to important medications and receive optimal therapies [23].

The present SR did not identify any studies to be included and can therefore be considered an empty review. The definition of empty review is "having no eligible studies retrieved or located by the review authors" [24]. Different reasons for an empty review have been proposed. One is that a subject/research area might be new and therefore not researched. Another is that the topic is very specific and no studies can to be found. A third reason is the use of overly stringent inclusion criteria [25]. In addition, publication bias, i.e. more publications of studies with positive findings compared to studies with no or negative results, could contribute to empty reviews [26, 27].

Possible limitations of this SR could be that the outcome measure (pain after dental treatment assessed by the child patient) was too narrow. However, patient-reported and patient-centred outcomes are essential in clinical research and based on the definition of pain being a subjective experience, the used outcome measure is highly relevant [28, 29]. This is in accordance with the COMET Handbook [30], which also states the importance of outcome measurement being appropriate and central for the key participants, including patients. Notably, no studies were excluded because of this inclusion criterion.

The definition of the population could also be discussed, as a limitation of this SR. As the review aimed to look at children and adolescents the age group, 3 to 19 years of age, was

chosen in order to find as many publications as possible and to ensure that the whole teenage period was included. The literature search identified a considerable number of papers; 3,963 records of which 216 were read in full text. The majority of the excluded publications (S2) did not provide data for children or adolescents (i.e. the population intended for this review). Based on this, it is not likely that the inclusion criteria were too stringent. Instead, the problem comes down to the fact there are too few studies on postoperative pain management in children and adolescents. This is in accordance with the findings in the Cochrane review on pre-emptive administration of oral analgesics in young patients aged up to 17 years that identified 1,691 records and was able to include only two studies in a quantitative synthesis [8].

Empty systematic reviews are important to report as they highlight research gaps and indicate the state of research evidence at a particular point in time [19]; they may also serve as a guide for researchers and/or funders towards novel areas and future original research that needs to be undertaken [25, 31]. There is also a risk of publication bias affecting decision-making in health care if not publishing empty systematic reviews. This problem is acknowledged by the WMA Declaration of Helsinki that raises the ethical obligation for researchers, authors, and editors etc. to publish and disseminate negative and inconclusive as well as positive results or research [32].

Within the field of paediatric dentistry, Mejàre et al. [33] identified and mapped a large number of knowledge gaps and concluded that there was an urgent need for good-quality primary clinical research in most clinically relevant domains. One domain pointed out was the "use of analgesics for the delivery of dental care" [33]. Also, the SBU database (Swedish Agency for Health Technology Assessment and Assessment of Social Services), serving as a repository for the UK Database of Uncertainties about the Effects of Treatments (DUETs) [34] has identified a need for a systematic review on postoperative pain relief for oral procedures in children and adolescents. The present SR meets this need [35] and thereby contributes to assembling the puzzle of research strategies related to pain reduction in conjunction with dental treatment in children and adolescents.

This HTA and SR points to a significant problem in that pharmacological substances are used in clinical practice without having been scrutinized. Also, the weighting of possible effects and side effects of the drugs or the intervention is lacking. Therefore, it is important to disseminate and discuss the results. Mainly two pharmacological substances, paracetamol (acetaminophen) and ibuprofen, have been suggested for treatment of pain resulting from dental treatment [5, 8]. Paracetamol is considered a very safe drug and is used for the treatment of pain and fever. However, there is a risk of toxicity from overdose or from underlying patient conditions that might be affected by the drug: for instance, dehydration, malnutrition, or concomitant use of other medications [36]. It is known that NSAIDs can precipitate asthma in sensitive individuals, although this is uncommon (less than 10%). Individuals sensitive to NSAIDs are often also sensitive to other unrelated COX inhibitors: for example, paracetamol [37]. This association between paracetamol and asthma is still under debate, since the evidence is inconclusive [38]. Ibuprofen, an NSAID, is also considered a safe pharmacological substance, and alongside paracetamol it is recommended as an antipyretic and analgesic from an early age [38]. However, there have been reported side effects in children from the use of ibuprofen, even though a clear association between ibuprofen and, for example, asthma or Reye's syndrome, has not been established [39, 40]. This calls for caution and highlights the importance of using only recommended standard doses of oral analgesics, based on weight and age [34]. Based on this knowledge, an empty systematic review is even more important. The lack of scientific evidence makes it impossible to construct any guidelines on the general administration of oral analgesics to prevent postoperative pain. Instead, all administration must be individually tailored and founded on a risk assessment that considers the type of dental

treatment, the patient's medical status, previous pain experiences, and the patient's subjective point of views.

## Conclusions

As no studies meeting the inclusion criteria were identified, it was not possible to find any scientific support for the effects, nor provide any support for rejection, of postoperatively administered oral analgesics for the prevention or reduction of pain after dental treatment in children and adolescents. Thus, it is not possible to formulate clinical guidelines on this issue solely based on scientific evidence. There is an urgent need for further well-designed studies on how to prevent pain after dental treatment. This empty systematic review serves as an important starting point for research in this area.

## Supporting information

**S1 PRISMA. PRISMA 2009 checklist.**
(DOCX)

**S1 File. Search strategies.**
(DOCX)

**S2 File. Characteristics of excluded studies.** List of excluded full text papers.
(DOCX)

**S1 Data.**
(PDF)

## Author Contributions

**Conceptualization:** Henrik Berlin, Gunilla Klingberg.

**Formal analysis:** Henrik Berlin, Karin Ridell, Susanne Brogårdh-Roth, Elisabeth Lager, Thomas List, Thomas Davidson, Gunilla Klingberg.

**Funding acquisition:** Gunilla Klingberg.

**Investigation:** Henrik Berlin, Martina Vall, Elisabeth Bergenäs, Karin Ridell, Susanne Brogårdh-Roth, Elisabeth Lager, Thomas List, Thomas Davidson, Gunilla Klingberg.

**Methodology:** Henrik Berlin.

**Project administration:** Henrik Berlin.

**Writing – original draft:** Henrik Berlin.

**Writing – review & editing:** Henrik Berlin, Martina Vall, Elisabeth Bergenäs, Karin Ridell, Susanne Brogårdh-Roth, Elisabeth Lager, Thomas List, Thomas Davidson, Gunilla Klingberg.

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
