## [Decision Letter · Decision Letter 0]

23 Aug 2019

PONE-D-19-18628

Effects and cost-effectiveness of postoperative oral analgesics for additional postoperative pain relief in children and adolescents undergoing dental treatment: Health technology assessment including a systematic review

PLOS ONE

Dear Dr Berlin,

Thank you for submitting your manuscript to PLOS ONE. After careful consideration, we feel that it has merit but does not fully meet PLOS ONE’s publication criteria as it currently stands. Therefore, we invite you to submit a revised version of the manuscript that addresses the points raised during the review process.

We would appreciate receiving your revised manuscript by 30 days. To enhance the reproducibility of your results, we recommend that if applicable you deposit your laboratory protocols in protocols.io, where a protocol can be assigned its own identifier (DOI) such that it can be cited independently in the future. For instructions see: http://journals.plos.org/plosone/s/submission-guidelines#loc-laboratory-protocols

We look forward to receiving your revised manuscript.

Kind regards,

Federico Bilotta

Academic Editor

PLOS ONE

Journal Requirements:

"This study was supported by research funds from Oral Health Related Research by Region Skåne (Odontologisk Forskning i Region Skåne, OFRS 569491), Sweden".

"The funders had no role in study design, data collection and analysis, decision to

publish, or preparation of the manuscript".

3. Please include a copy of Table 1 which you refer to in your text on page 6.

Reviewers' comments:

Reviewer's Responses to Questions

**Comments to the Author**

1. Is the manuscript technically sound, and do the data support the conclusions?

Reviewer #1: No

Reviewer #2: Yes

2. Has the statistical analysis been performed appropriately and rigorously? 

Reviewer #1: No

Reviewer #2: N/A

3. Have the authors made all data underlying the findings in their manuscript fully available?

Reviewer #1: No

Reviewer #2: Yes

4. Is the manuscript presented in an intelligible fashion and written in standard English?

Reviewer #1: No

Reviewer #2: Yes

5. Review Comments to the Author

Reviewer #1: The preparation of Systematic review was erroneous.

Several lacunae were noted and authors should discuss with someone who has prepared SRs before, especially those who have worked with the Cochrane Collaboration.

Outcomes are not well defined and the PICO statement is incorrectly prepared.

Professional help should have been taken for preparation of Search Strategies.

Reviewer #2: Manuscript Number: PONE-D-19-18628

Title: Effects and cost-effectiveness of postoperative oral analgesics for additional postoperative pain relief in children and adolescents undergoing dental treatment: Health technology assessment including a systematic review

Submitted to:

PLOS ONE

1. ABSTRACT.

- Abstract appropriately summarize the manuscript.

- There aren´t discrepancies between the Abstract and the remainder of the manuscript.

- The Abstract can be understood without reading the manuscript.

2. BACKGROUND/AIM

- The Introduction is concise.

- The purpose of the study is well defined.

- The authors provide a rationale for performing the study based on a review of the medical literature with an appropriate length.

- This manuscript is Original Research, with a well-defined hypothesis.

3. MATERIALS AND METHODS

- Please, add a PRISMA statement checklist.

- Please, add an AMSTAR checklist.

4. RESULTS

- The results are clearly explained.

- The results are reasonable and expected.

- There aren´t results introduced that are not preceded by an appropriate discussion in the Methods section.

5. DISCUSSION

- The discussion is concise.

- Their research question was answered.

- Please, define Limitations of the study.

- Authors’ conclusions are justified by the results found in the study.

6. FIGURES

- Figure is appropriate and it is appropriately labeled.

- Adequately show the important results.

7. TABLES (Supl)

- Appropriately describe the results.

8. REFERENCES

- The reference list follows the format for the journal.

6. PLOS authors have the option to publish the peer review history of their article (what does this mean?). If published, this will include your full peer review and any attached files.

Reviewer #1: No

Reviewer #2: Yes: Rafael Badenes

---

## [Author Response · Author response to Decision Letter 0]

18 Sep 2019

Reviewers' and editor's comments has been responded to in the document "Response to Reviewers", that has been attached to present re-submission. 

We have considered the Journal Requirements and comments and critique from the reviewers. We have made changes accordingly, but also have following comments:

JOURNAL REQUIREMENTS:

[Authors’ comments]: We have updated the file names according to the links above, and adjusted the manuscript to meet PLOS ONE’s style requirements.

2. Thank you for stating the following in the Acknowledgments Section of your manuscript: "This study was supported by research funds from Oral Health Related Research by Region Skåne (Odontologisk Forskning i Region Skåne, OFRS 569491), Sweden". We note that you have provided funding information that is not currently declared in your Funding Statement. However, funding information should not appear in the Acknowledgments section or other areas of your manuscript. We will only publish funding information present in the Funding Statement section of the online submission form. Please remove any funding-related text from the manuscript and let us know how you would like to update your Funding Statement. Currently, your Funding Statement reads as follows:

"The funders had no role in study design, data collection and analysis, decision to publish, or preparation of the manuscript".

[Authors’ comments]: The funding information is removed from the manuscript. For the Funding section, we would like to update the text with the following:

“This study was supported by research funds from Oral Health Related Research by Region Skåne (Odontologisk Forskning i Region Skåne, OFRS 569491), Sweden. 

3. Please include a copy of Table 1 which you refer to in your text on page 6.

[Authors’ comments]: Table 1 is now included in present re-submission.

REVIEWERS’ COMMENTS:

Reviewer's Responses to Questions

Comments to the Author

1. Is the manuscript technically sound, and do the data support the conclusions?

Reviewer #1: No

[Authors’ comments]: The manuscript has been prepared according to common principles for systematic reviews (SR) and health technology assessments (HTA). We have followed the guidelines for SRs, from SBU (Swedish Agency for Health Technology Assessment and Assessment of Social Services) (Line 393-397, page 16). SBU is in the same network as Cochrane Sweden, which is a part of the Cochrane global community. This SR also follows the PRISMA-statement (Line 158-159, page 6)

Reviewer #2: Yes

2. Has the statistical analysis been performed appropriately and rigorously?

Reviewer #1: No

[Authors’ comments]: Since this SR came out empty, no statistical analysis could be performed. This question is to our understanding therefore not applicable.

Reviewer #2: N/A

3. Have the authors made all data underlying the findings in their manuscript fully available?

Reviewer #1: No

[Authors’ comments]: Reviewer #1 has not given any thorough review on what is missing. We therefore find it difficult to revise the manuscript, without this detailed information. 

Reviewer #2: Yes

4. Is the manuscript presented in an intelligible fashion and written in standard English? 

Reviewer #1: No

[Authors’ comments]: Language editing was undertaken before submitting the manuscript. This was done by an English speaking professional English Language Consultant with long experience in scientific language.

Reviewer #2: Yes

5. Review Comments to the Author

Reviewer #1: 

• The preparation of Systematic Review was erroneous. Several lacunae were noted and authors should discuss with someone who has prepared SRs before, especially those who have worked with the Cochrane Collaboration.

[Authors’ comments]: This systematic review has been performed according to the guidelines from the Swedish Agency for Health Technology Assessment and Assessment of Social Services (SBU), which is a Swedish governmental agency focusing on assessing and evaluating methods in use in healthcare and social services. It is one of the world’s oldest HTA agencies, collaborating with Cochrane Sweden. We have referred to this in the manuscript (Line 175-178, side 7-8; ref #16 (Swedish Agency for Health Technology Assessment and Assessment of Social Services (SBU). Assessment of methods in health care—A handbook. Stockholm. 2018. Available: https://www.sbu.se/contentassets/76adf07e270c48efaf67e3b560b7c59c/eng_metodboken.pdf)). The PRISMA-statement has been followed as well (Line 159, side 6; ref #14 (Moher D, Liberati A, Tetzlaff J, Altman DG, The PG. Preferred reporting items for systematic reviews and meta-analyses: the PRISMA statement. PLoS Med. 2009;6(7):e1000097. doi: 10.1371/journal.pmed.1000097. PMID: 19621072)). 

• Outcomes are not well defined and the PICO statement is incorrectly prepared.

[Authors’ comments]: We believe that the PICO is formulated correctly. We have defined the Population (Line 104, page 4), Interventions (Line 108, page 4 and Line 109-114, page 5), Control (line 116-124, page 5), and Outcome Measures (line 126-130, page 5). The SR has also been registered on PROSPERO (https://www.crd.york.ac.uk/PROSPERO/). 

• Professional help should have been taken for preparation of Search Strategies.

[Authors’ comments]: This has been made (line 155-156, page 6). Two librarians specialized in informatics performed the search in all five databases, used in this SR. 

Reviewer #2 (only the bullet points requiring revision): 

• Please, add a PRISMA statement checklist.

[Authors’ comments]: PRISMA checklist was added in the first submission. We have however added it again with the current re-submission. 

• Please, add an AMSTAR checklist.

[Authors’ comments]: This is added with the current re-submission.

• Please, define Limitations of the study.

[Authors’ comments]: Limitations of our study are mentioned, see Line 268-286, page 11-12. However we have inserted a phrase in the beginning of Line 268, page 11, and at the end of the first sentence on line 275, page 11, to make it clearer that limitations are discussed. “Possible limitations of this SR could be that the outcome measure (pain after dental treatment assessed by the child patient) was too narrow.”, and “The definition of the population could also be discussed, as a limitation of this SR.”

---

## [Decision Letter · Decision Letter 1]

12 Dec 2019

Effects and cost-effectiveness of postoperative oral analgesics for additional postoperative pain relief in children and adolescents undergoing dental treatment: Health technology assessment including a systematic review

PONE-D-19-18628R1

Dear Dr. Berlin,

We are pleased to inform you that your manuscript has been judged scientifically suitable for publication and will be formally accepted for publication once it complies with all outstanding technical requirements.

With kind regards,

Federico Bilotta

Academic Editor

PLOS ONE

Additional Editor Comments (optional):

PONE-D-19-18628R1

In this SR, the Authors conducted a systematic review (SR) and health technology assessment (HTA) of oral analgesics administered after dental treatment to prevent postoperative pain in children and adolescents aged 3-19 years.

A PICO-protocol was constructed and registered in PROSPERO (CRD42017075589). Searches were conducted in PubMed, Cochrane, Scopus, Cinahl, and EMBASE, November 2018. The researchers (reading in pairs) assessed identified studies independently, according to the defined inclusion and exclusion criteria, following the PRISMA-statement.

A total of 3,963 scientific papers were identified, whereof 216 read in full text. None met the inclusion criteria, leading to an empty SR. Ethical issues were identified related to the recognized knowledge gap in terms of challenges to conduct studies that are well-designed from methodological as well as ethical perspectives.

The Authors concluded that, there is no scientific support for the use or rejection of oral analgesics administered after dental treatment in order to prevent or reduce postoperative pain. Thus, no guidelines can be formulated on this issue based solely on scientific evidence. Well-designed studies on how to prevent pain from developing after dental treatment in children and adolescents is urgently needed.

The study is interesting and well conducted. The manuscript is well written and informative

Comments

Reviewer 1: No competing interests. Accept

Reviewer 2: The authors have satisfactorily responded to all my questions and made the necessary changes to the manuscript. Accept

Reviewers' comments:

Reviewer's Responses to Questions

**Comments to the Author**

1. If the authors have adequately addressed your comments raised in a previous round of review and you feel that this manuscript is now acceptable for publication, you may indicate that here to bypass the “Comments to the Author” section, enter your conflict of interest statement in the “Confidential to Editor” section, and submit your "Accept" recommendation.

Reviewer #1: All comments have been addressed

Reviewer #2: All comments have been addressed

2. Is the manuscript technically sound, and do the data support the conclusions?

Reviewer #1: Yes

Reviewer #2: Partly

3. Has the statistical analysis been performed appropriately and rigorously? 

Reviewer #1: I Don't Know

Reviewer #2: N/A

4. Have the authors made all data underlying the findings in their manuscript fully available?

Reviewer #1: Yes

Reviewer #2: Yes

5. Is the manuscript presented in an intelligible fashion and written in standard English?

Reviewer #1: Yes

Reviewer #2: Yes

6. Review Comments to the Author

Reviewer #1: (No Response)

Reviewer #2: (No Response)

7. PLOS authors have the option to publish the peer review history of their article (what does this mean?). If published, this will include your full peer review and any attached files.

Reviewer #1: No

Reviewer #2: Yes: Rafael Badenes

---

## [Editor Report · Acceptance letter]

17 Dec 2019

PONE-D-19-18628R1 

Effects and cost-effectiveness of postoperative oral analgesics for additional postoperative pain relief in children and adolescents undergoing dental treatment: Health technology assessment including a systematic review 

Dear Dr. Berlin:

I am pleased to inform you that your manuscript has been deemed suitable for publication in PLOS ONE. Congratulations! Your manuscript is now with our production department. 

With kind regards,

on behalf of

Dr. Federico Bilotta 

Academic Editor

PLOS ONE